# Enhancing Consistency-Based Image Generation via Adversarially-Trained Classification and Energy-Based Discrimination

**Shelly Golan**
Technion
shelly.golan@cs.technion.ac.il

**Roy Ganz**
Technion
ganz@campus.technion.ac.il

**Michael Elad**
Technion
elad@cs.technion.ac.il

## Abstract

The recently introduced Consistency models pose an efficient alternative to diffusion algorithms, enabling rapid and good quality image synthesis. These methods overcome the slowness of diffusion models by directly mapping noise to data, while maintaining a (relatively) simpler training. Consistency models enable a fast one- or few-step generation, but they typically fall somewhat short in sample quality when compared to their diffusion origins. In this work we propose a novel and highly effective technique for post-processing Consistency-based generated images, enhancing their perceptual quality. Our approach utilizes a joint classifier-discriminator model, in which both portions are trained adversarially. While the classifier aims to grade an image based on its assignment to a designated class, the discriminator portion of the very same network leverages the softmax values to assess the proximity of the input image to the targeted data manifold, thereby serving as an Energy-based Model. By employing example-specific projected gradient iterations under the guidance of this joint machine, we refine synthesized images and achieve an improved FID scores on the ImageNet 64x64 dataset for both Consistency-Training and Consistency-Distillation techniques.

## 1 Introduction

Diffusion models, also known as score-based generative models, [13], [24], [27], [28], [29] have set new benchmarks in multiple fields, the prime of which is image generation [4], [22]. A central aspect of these models is the iterative sampling process that gradually eliminates noise from an initial random vector, this way carving fair samples from a prior distribution. By controlling the depth of such processes, these methods offer a flexible balance between computational effort and sample quality. However, compared to single-step generative models like GANs [9], the iterative generation procedure of diffusion models typically requires 10–2000 times more compute for sample generation [29], [28], [13], causing slow inference and limited real-time applications. Addressing this shortcoming, Consistency models [26] have been developed to expedite the sampling process, offering a faster alternative with one or two-step data generation. Unfortunately, their reliance on an intricate distillation process tends to limit the ultimate quality of the generated images.

Recent work that aims to remedy this performance gap integrates Consistency Models [26] with Generative Adversarial Networks, so as to tackle common GAN issues like mode collapse and unstable training process, thereby enhancing the diversity and stability of generated images [20].

38th Conference on Neural Information Processing Systems (NeurIPS 2024).

Adversarial Consistency Training (ACT) [17] further refines this approach by incorporating a discriminator within the Consistency training framework, improving the generative model's quality by fine-tuning the image generation process for more realistic outputs. These methods offer new ways to overcome limitations in image fidelity, but require long and computationally expensive training.

In this work we propose a novel and highly effective technique for post-processing Consistency-based generated images, enhancing their perceptual quality. Our approach is based on a joint classifier-discriminator neural network, in which both functionalities are trained adversarially.

The adversarial approach is critical in our training process, as we will further explain in section 3. We rely on past research that has shown that deep neural networks can be easily deceived by adversarial attacks [10], [18], [30]. Therefore, numerous strategies have been developed to make these networks more robust, with adversarial training becoming one of the most favored approaches [21], [10]. During the analysis of such classifiers, a notable phenomenon was discovered: perceptually aligned gradients [31], [5]. This trait implies that adjustments to an image, guided by the robust classifier, result in changes that are visually consistent with the intended classification.

Our method utilizes the above phenomenon and iteratively modifies the images created by Consistency-models using a joint robust classifier-discriminator. While the classifier aims to maximize the conditional probability of a target class, the discriminator portion of the very same network uses the softmax values to maximize the energy difference between synthesized and observed data, functioning as an Energy-based Model [19].

In order to refine synthesized images, we employ example-specific projected gradient iterations under the guidance of this joint machine. The process is controlled by an image-dependent early-stopping mechanism, that allows us to selectively refine images that diverge from the observed data distribution, ensuring that those already close to it remain unaffected. To summarize, our contributions are as follows:

1. We show significant quantitative and qualitative improvements in the perceptual quality of synthesized images for both Consistency-Training and Consistency-distillation, with an FID boost of $27.48\%$ and $20.96\%$, respectively.

2. We demonstrate a remarkable capability of adversarial robust classifiers: Serving as a joint classifier-discriminator informed of both synthesized and observed data distributions. This fusion leads to enhanced effectiveness of projected-gradient steps, leading to an additional improvement of $11.2\%$ in FID results, over a classifier boosting alone [6].

3. While this work focuses on boosting Consistency based image generation, the proposed approach is general and applicable to any other generative model. While this is beyond the scope of the current work, we do present supporting preliminary such results.

## 2 Background

### 2.1 Diffusion Models and Consistency

Diffusion models [29], [14] generate data (e.g., images) by starting from a plain random vector and manipulating it progressively by sequential denoising steps and noise perturbations. More specifically, let $p_0(x)$ denote the Probability Density Function (PDF) of true images. Diffusion models introduce a forward path, in which $p_0(x)$ is gradually diffused over the time interval $0 \leq t \leq T$ to a canonical Gaussian, $p_T(x) = \mathcal{N}(0, I)$. This is formulated via a Stochastic Differential Equation (SDE) that describes a continuous noise-mixing process,

$$dx_t = \mu(x_t, t)dt + \sigma(t)dw_t. \tag{1}$$

Here $T > 0$ is a fixed constant, $\mu(\cdot, \cdot)$ and $\sigma(\cdot)$ are the drift and diffusion coefficients respectively, and $\{w_t\}_{t \in [0,T]}$ denotes the standard Brownian motion. We denote the distribution of $x_t$ as $p_t(x)$, where $p_t(x)$ is a $\sigma(t)$-blurred version of $p_0(x)$ due to the additive noise in $x_t$.

Diffusion models are all about designing the reverse process from $t = T$ to $t = 0$, in which noise is filtered gradually to produce high quality images that serve as fair samples from $p_0(x)$. While the original diffusion techniques [13] offer an SDE-based reverse path, the more recent and more effective alternatives (e.g. DDIM [25]) remove the intermediate stochasticity and offer an ODE-based

reverse process of the form

$$dx_t = \left[ \mu(x_t, t)dt - \frac{1}{2}\sigma(t)^2 \nabla_{x_t} log p_t(x_t) \right] dt. \qquad (2)$$

Notice the appearance of the Score-Function $\nabla_{x_t} log p_t(x_t)$ in this flow, which is approximated by an image denoiser.

In practice, Diffusion Models are an Euler-Maruyama discretization [16] of the above equation, resulting with a chain of iterations that starts with a random canonical Gaussian noise $x_T$, and gradually moves $x_t$ towards $x_0$ by steps of denoising that follow the above equation. The resulting $x_0$ can be considered as a fair sample from $p_0(x)$, assuming that the denoiser is well-trained.

Consistency models [26] are heavily inspired by the theory described above, aiming to preserve the end-to-end behavior that takes us from $x_T$ to $x_0$, but while expediting the process to a single (or a few) step. Given a solution trajectory $\{x_t\}_{t \in [0,T]}$ of the above reverse equation, the Consistency function is the trained network $f : (x_t, t) \to x_0$. This function should have the property of self-consistency: For two temporal points $t, s \in [0, T]$ and their corresponding stages in the same trajectory, $x_t$ and $x_s$, the outputs should be the same $f(x_t, t) = f(x_s, s) = x_0$, thus explaining the name "Consistency". Such models can be trained in two main ways [26]. The first method (CD) involves distilling knowledge from a pre-trained diffusion model, allowing it to learn the distribution directly from the more computationally intensive yet well-performing diffusion process. The second method (CT) trains the function $f(\cdot, \cdot)$ directly via a tailored loss, without the need for a pre-existing diffusion model.

## 2.2  Boosting Synthesis via Robust Classifier (BIGROC)

Consider a goal of refining generated images, so as to improve their perceptual quality. BIGROC [6] offers such a post-processing mechanism, by leveraging a robust classifier.

Let $f_\theta : \mathbb{R}^d \to \mathbb{R}^C$ be a classifier that maps images in $\mathbb{R}^d$ to scores for $C$ classes, also known as logits, and denote it's $c^{\text{th}}$ output by $f_\theta^c(x)$. These logits aim to have a probabilistic interpretation, being related to the estimated posterior probability $p_\theta(c|x)$. Starting with an arbitrary image $x$ and taking gradient steps $\nabla_x f_\theta^c(x)$ in order to maximize $f_\theta^c(x)$, one would expect this process to produce an altered image that better resembles the targeted class $c$. However, when using a plain classifier, even if it is very well performing, its gradients behave as a perceptually meaningless noise that is unrelated to the image content. This implies that, while the resulting image would tend to be classified to $c$, its visual appearance would be hardly changed from the original image $x$. This phenomenon is well-known, giving rise to the wide topic of adversarial attacks on neural networks [21], [9].

Past research has shown that, as opposed to the above, robust classifiers produce Perceptually Aligned Gradients (PAG) [31, 15, 1, 8, 7]. These gradients are a trait of adversarially trained models, their content being consistent with the human visual perception [31], [5]. This implies that when used within the gradient steps mentioned above that aim to pull $x$ towards class $c$, robust models are expected to yield meaningful features aligned with the target class, thus modifying $x$ in a perceptually convincing way.

Harnessing the above knowledge, the BIGROC algorithm [6] proposes an iterative process for boosting the perceptual quality of synthesized images, assuming that they are classifiable via $f_\theta(x)$. This is obtained by modifying any generated image $x$ so as to maximize the posterior probability of a given target class $\hat{c}$, i.e, $p_\theta(\hat{c}|x)$, where $p_\theta$ is modeled by an adversarially trained classifier and $\hat{c}$ stands for the estimated class the image belongs to. The outcome of this process is a modified version of $x$ that is more aligned with the class $\hat{c}$ from a human point of view, thus having a better perceptual quality.

While there are various techniques for generating adversarial examples, the one used in BIGROC is Targeted Projected Gradient Descent [21] – a deterministic iterative process that operates as described in Algorithm 1, $\Pi_\varepsilon$ in this algorithm stands for the projection operator that ensures that the resulting image is $\varepsilon$-close to the starting image, and $\ell$ is the loss function (e.g., cross-entropy) that defines how the distance between the classifier output and the target label is computed. For simplicity, we can omit the loss and use the gradient of the softmax output of the classifier directly,

$$\delta_{t+1} = \Pi_\varepsilon(\delta_t - \nabla_\delta f_\theta^{\hat{c}}(x + \delta_t)). \qquad (3)$$

We would like to draw the readers attention to the fact that BIGROC [6] is totally unaware of the generated images' distribution, which it aims to improve. This brings us naturally towards the proposed strategy in this paper.

---

**Algorithm 1** Targeted Projected Gradient Descent (PGD)

---

**Input:**
- Robust Classifier $f_\theta(x)$,
- Input image $x$, and
- Algorithm's parameters: radius $\varepsilon$, step-size $\alpha$, no. of iterations $T$, and loss function $\ell$.

Set $\delta_0 \leftarrow 0$.
Set $\hat{c} = f_\theta(x)$ or get an external assignment for the class.
**for** $t$ from 0 to $T$ **do**
    $\delta_{t+1} = \Pi_\varepsilon(\delta_t - \alpha\nabla_\delta\ell(f_\theta(x + \delta_t), \hat{c}))$.
**end for**
**Output:** $x_{adv} \leftarrow x + \delta_T$

---

## 2.3 Energy-Based Models

The core idea behind Energy-Based Models (EBM) [19] is turning an arbitrary function into a probability distribution. Consider a learned energy function $E_\theta(x) : \mathbb{R}^d \to \mathbb{R}$, where $x \in \mathbb{R}^d$ is an input image, and it's output is a scalar value in the range $(-\infty, +\infty)$. This neural network induces a PDF $p_\phi(x)$ that has a Gibbs form:

$$p_\theta(x) = \frac{exp(-E_\theta(x))}{Z_\theta}. \tag{4}$$

Clearly, this function assigns a probability greater than zero to any input. $Z_\theta$ is known as the Partition Function – a normalization factor that ensures that the density integrates to 1. The negative sign in the exponent implies that low energy values are assigned to highly probable images, while high energy refers to poor looking images.

EBMs are a family of algorithms that aim to learn $E_\theta(x)$ such that the resulting $p_\theta(x)$ aligns with the probability density function of the given training set. Given a learned EBM, image synthesis can be obtained by iteratively climbing the function $p_\theta(x)$ using Stochastic Gradient Langevin Dynamics (SGLD) [32]. PGD, as described above, offers an appealing deterministic alternative to SGLD, by leveraging a loss-function and an additional $\varepsilon$-ball constraint in it's computation.

Energy-based models and image classifiers can be shown to have common grounds [11], as detailed in Appendix A. Assuming we have a classifier $f_\theta(x) : \mathbb{R}^d \to \mathbb{R}^C$ that maps images to scores for $C$ classes, the probability of the $c^{th}$ label is represented using the Softmax function,

$$p_\theta(c|x) = \frac{exp(f_\theta^c(x))}{\sum_{\tilde{c}=1}^{C} exp(f_\theta^{\tilde{c}}(x))}, \tag{5}$$

The normalizing denominator in the above equation can be turned into a candidate energy function for an EBM,

$$E_\theta(x) = -log\left[\sum_{c=1}^{C} exp(f_\theta^c(x))\right]. \tag{6}$$

In words, the exponent of this energy is the sum of the logits' exponentials. And so, while the classifier's assignment decision, $p_\theta(c|x)$, is independent of the induced energy, $E_\theta(x)$, this value can be interpreted as the entity that defines the PDF $p_\theta(x)$ that aims to match a given image distribution. In this work we leverage this possibility, and modify it, as discussed hereafter.

## 3 Proposed Approach

In this work we propose a method to enhance the perceptual quality of images generated by Consistency-based techniques [26]. Our approach utilizes a robust joint classifier-discriminator to

guide synthesized images to better resemble real images. In section 3.1, we explore this method's training objectives and technical details. The post-processing procedure, described in section 3.2, uses the guidance of the joint model in an iterative algorithm to refine the realism and visual appeal of synthesized images. In this process, the synthesized images, originating from the generated distribution $p_\theta$, are modified to align with the real images distribution $p_{data}$.

## 3.1 Training a Joint Classifier-Discriminator

Our method aims to leverage the Perceptually Aligned Gradients (PAG) property [31], and further extend and improve refinement capabilities of models that possess this trait by introducing the model with the distribution of both $p_\theta$ and $p_{data}$. To this end, we empower the classifier model by incorporating a robust discriminator within it, so as to improve guidance performance significantly with this joint model. This dual-training approach allows the proposed model to better differentiate and understand the unique features of both real and synthetic images.

### 3.1.1 Proposed Training objective

While Energy-Based Models (EBMs) [19] may align perfectly with our task of improving image quality by modifying out-of-distribution samples to align with the real data manifold, several challenges impact their practical application. These include the sensitivity of EBM training to the choice of the energy function used, leading often to training instability and poor generalization performance. EBMs necessarily rely on computationally complex sampling techniques (e.g. MCMC), which add to their overall complexity and instability. More on these matters is brought in Appendix B.

In this work we propose a joint classifier-discriminator model that overcomes many of the EBM problems mentioned above. Our model aims to guide generated samples to be more realistic and emphasize visual features aligned with the target class. To do so, we propose a revised interpretation of the adversarial training objective and a modified view of the energy that serves the EBM part of our model. More specifically, the loss function proposed contains adversarial variants of both Cross-Entropy and Binary-Cross-Entropy components, each targeting specific aspects of the training as explained further below.

**Discrimination loss:** The Binary-Cross-Entropy (BCE) loss we suggest is formulated as follows:
$$\mathcal{L}_{BCE}(\theta) = \mathbb{E}_{x \sim p_{data}}[\ell_{BCE}(\hat{x}, c)] + \mathbb{E}_{x \sim p_\theta}[\ell_{BCE}(\hat{x}, c)], \tag{7}$$
where we define $\ell_{BCE}(x, c)$ for an image $x$ from class $c$ via
$$\ell_{BCE}(x, c) = \begin{cases} -log(\sigma(f_\theta^c(x))) & x \sim p_{data} \\ -log(1 - \sigma(f_\theta^c(x))) & x \sim p_\theta. \end{cases} \tag{8}$$

This loss admits values in the range $[0, \infty)$. For real images, the value $f_\theta^c(x)$ should tend to be high, while for generated ones it should strive to low values. As such, this ingredient behaves as an energy function that serves our discriminator.

In these equations, $\hat{x}$ represents the adversarially perturbed version of $x$, $\sigma$ is the sigmoid function, $f_\theta$ is a classifier and $c$ is the class to which the sample $x$ belongs. In this setting, $\hat{x}$ derived from $x \sim p_{data}$ serves as an adversarial example designed to deceive the model by pushing it to be "less real". Conversely, $\hat{x}$ derived from $x \sim p_\theta$ represents a synthetic image that has been created by the Consistency model, and was adjusted to appear more realistic. As we show in our ablation study, these adversarial examples are essential to the training process.

Note that in both attacks, the energy function of the classifier, as given in Equation 6, is *not* the one used for the PGD computation. Rather, we use only the $c$ logit magnitude as an energy proxy. Similarly to EBMs, we suggest assigning high probability values to real images and low probability values to fake ones. Further discussion about the relation between the suggested adversarial training and EBMs can be found in Appendix C.

**Classification loss:** The commonly used Cross-Entropy loss is formulated as follows:
$$\mathcal{L}_{CE}(\theta) = \mathbb{E}_{x \sim p_{data}}[\ell_{CE}(\hat{x}, c)] + \mathbb{E}_{x \sim p_\theta}[\ell_{CE}(\hat{x}, c)], \tag{9}$$
where $\ell_{CE}(x, c)$ for an image $x$ from class $c$ is defined by a soft-max operation,
$$\ell_{CE}(x, c) = -log \frac{exp(f_\theta^c(x))}{\sum_{c'=1}^{C} exp(f_\theta^{c'}(x))}, \tag{10}$$

which leads to values in the range $[0, \infty)$, 0 for high classification probability and large values for indecisive classification. Here, $\hat{x}$ are the adversarially perturbed samples as described in the paragraph above, in the context of the BCE loss.[1] This loss ensures that the classifier can accurately classify real images to their correct classes, despite any potential adversarial modifications, thereby reinforcing the model's resilience to adversarial attacks and maintaining high classification accuracy on real data. Similarly to the BCE loss, this term calculates the CE loss for both the true and the generated images, maintaining the model's ability to classify generated images correctly.

The overall loss is simply an addition of the above two expressions, $\mathcal{L}(\theta) = \mathcal{L}_{CE}(\theta) + \mathcal{L}_{BCE}(\theta)$. This formulation ensures that the model learns to distinguish effectively between real and generated images, as well as classifying them correctly, enhancing its ability to deal with adversarially altered images and to generalize well across different types of image manipulations. More importantly, due to the adversarial training, the learned model has meaningful gradients (*i.e.,* PAG [31], [5]), which are about to play a vital role in our inference process of pushing Consistency generated images to their improved versions. As our training strategy considers both classification and discrimination losses, these gradients carry both improvements geared by class adjustments and a migration from $p_\theta$ to $p_{data}$.

In our training process, as described above, we initialize the joint model with a robust RN50 network [21], while replacing its fully connected head with a randomly initialized one. As a consequence, we expect this network to inherit it's original classification robustness, which proves to be valuable for the overall image improvement obtained hereafter. Our training stands on the shoulders of this classifier, as we update the model parameters with the loss described above using only attacked (real and synthesized) images.

### 3.1.2 Gradual Adversarial Training

When computing the adversarial attacks for the above losses, we operate differently on real and synthesized images. A real image $x$ is modified by the PGD algorithm by maximizing the $\ell_{BCE}(x, c)$ loss, aiming to push it away from $p_{data}$. This is achieved by ascent directions that require a negative value of $\alpha$ in Algorithm 1. In contrast, attacks on the synthesized images aim to minimize $\ell_{BCE}(x, c)$ with a positive $\alpha$ in the PGD algorithm.

Referring to the later attacks on the synthesized images, $x \sim p_\theta$, PGD applies $T$ steps of gradient descent that explore a local area with a small $\varepsilon$ around the starting point of each input sample. We suggest to modify the activation of the PGD attack to a gradual form, using a large value of $\varepsilon$, as explained below. In forming the attacks mentioned above, our method aims to traverse the entire spectrum between fake and real samples, systematically eliminating any miss-classifications of fake images as reals ones by the model.

We propose a strategy of gradually increasing the number of attack steps $T$ throughout the training process. By extending the number of steps and initializing $\varepsilon$ to be relatively large, the PGD attack can move further from the initial starting points, eliminating closer misleading examples first, and then proceeding to more challenging adversarial examples. Moreover, as the model becomes more adept at countering simpler attacks, incrementally increasing the complexity of the attacks ensure that the training continues to challenge the model effectively.

To further improve our training process and ensure the model does not mistakenly classify high-quality but still fake samples as genuine ones, we implement an early stopping mechanism based on the model's probability outputs. This mechanism halts the PGD process when the model assigns a higher probability to a perturbed fake image being real than it does to a perturbed real image in the same batch. This mechanism prevents the model from believing that fake images are more "real" than the real ones, preserving its' reliability and discriminative power.

### 3.2 Sampling

As we move to inference, we aim to harness the PAG obtained by the adversarial training of the joint model, in order to boost the perceptual quality of Consistency generated images. We propose an

---

[1]One might claim that a different attack should be practiced here, so as to accommodate the classification task. Our experiments show that, while this option can be practiced, it is expected to mildly improve the results, while consuming substantial computational efforts.

iterative algorithm designed to adjust images $x \sim p_\theta$, by maximizing the probability that $x$ belongs to class $c$ and simultaneously minimizing the gap between $p_\theta$ and $p_{data}$. The choice of the loss function, $\ell$, is critical in this part, as will be shown in the ablation study. To utilize the gradients of the joint model we start by proposing the following expression that we minimize via the PGD:

$$\ell(x, c) = \ell_{CE}(x, c) - \ell_{BCE}(x, c). \tag{11}$$

Notice the marked difference between the above and the loss used for the training. This function operates only on incoming synthesized images $x$ and while knowing their designated class $c$. The rational in this expression has two pieces: A desire to minimize $\ell_{CE}(x, c)$ so as to align the image with its target class, just as practiced by BIGROC. In addition, $\ell_{BCE}(x, c)$ should strive for a high value, so as to make the image as realistic as possible, moving $x$ closer to $p_{data}$. We remind the reader that $\ell_{BCE}(x, c)$ here refers to synthesized images (see Equation 7), and thus a high value corresponds to high perceptual quality.

In practice, our tests indicate that the above loss tends to stray unstably, as $\ell_{BCE}(x, c)$ admits too high values, diverting from good quality outcomes. A remedy to this effect is the following alternative loss:

$$\ell(x, c) = \ell_{CE}(x, c) - \frac{1}{2}\left(\ell_{BCE}(x, c) - \ell_0\right)^2. \tag{12}$$

Here we aim to push $\ell_{BCE}(x, c)$ towards a reference value $\ell_0$, set in the training process to be

$$\ell_0 = \mathbb{E}_{x \sim p_{data}}[\ell_{BCE}(x, c)]. \tag{13}$$

In words, $\ell_0$ is the mean value of the BCE loss for real images. Minimizing the distance between $\ell_{BCE}(x, c)$ and $\ell_0$ pushes the modified image to be more "real", while also making sure that it does not admit a value above $\ell_0$, which represents an exaggerated quality.

The overall algorithm for boosting images $x \sim p_\theta$ is described in Algorithm 2.

---

**Algorithm 2** Boosting Images via PGD Guided by a Joint Classifier-Discriminator

**Input:**
- Robust Joint Classifier-Discriminator $f_\theta(x)$,
- Input image $x$ of a target class $c$, and
- Algorithm's parameters: Radius $\varepsilon$, Step-size $\alpha$, No. of iterations $T$, and loss function $\ell$.

**Output:** $x_{boosted} \leftarrow TargetedPGD(f_\theta, \ell, x, c, \varepsilon, \alpha, T)$.

---

An alternative approach is to sample using Stochastic Gradient Langevin Dynamics, similarly to [11], as described in Algorithm 3. To further investigate the properties of SGLD, we explored replacing PGD with SGLD during both inference and training. During training, we found that optimizing with SGLD lacks the robustness achieved with PGD and suffers from convergence difficulties. However, during inference, SGLD surpasses PGD, as shown in Table 3. The benefit of using SGLD during inference can be attributed to its ability to better explore multiple modes compared to methods like PGD due to it's stochastic nature.

---

**Algorithm 3** Boosting Images via SGLD Guided by a Joint Classifier-Discriminator

**Input:**
- Robust Joint Classifier-Discriminator $f_\theta(x)$,
- Input image $x$ of a target class $c$, and
- Algorithm's parameters: Step-size $\alpha$, No. of iterations $T$, noise level $\sigma$.

**for** $t$ from 1 to $T$ **do**
$\quad x_t = x_{t-1} - \alpha \cdot \nabla_{x_{t-1}} f_\theta(x_{t-1})[c] + \sigma \cdot \mathcal{N}(0, \mathbf{I}).$
**end for**
**Output:** $x_T$.

---

## 4  Experiments

In this section, we present an evaluation of the proposed boosting method. We begin by discussing the results obtained by our method compared to BIGROC [6]. Subsequently, we conduct an ablation study to further analyze various components of the model, as described in Section 3.

## 4.1 Experimental Settings

During training, we use the pre-trained robust ResNet-50 (RN50) model [21] as a backbone, with a randomly initialized classification head. We configure the parameters of the Projected Gradient Descent (PGD) attack for fake images with an $\varepsilon$ of 10.0 and a step size of 0.1, increasing the number of PGD steps every 1,000 training iterations to enhance adversarial robustness over time. For real images, the PGD attack is set with an $\varepsilon$ of 1.0, a step size of 0.25 and 4 steps. The model is trained on four NVIDIA GeForce RTX 3090 GPUs. The same configuration applies to the training of the Wide-ResNet-50-2 (Wide-RN) model. In the sampling process, we apply the loss function outlined in Equation 12 and the step size is set as 0.1. The amount of steps is adjusted according to the generative model we enhance; further technical details can be found in Appendix D.

## 4.2 Experimental Results

We analyze the perceptual-quality boosting performance using Fréchet Inception Distance [12], Inception Score [23], Precision and Recall. Table 1 summarizes our results on the ImageNet 64×64 dataset [3]. Initially, we present the original FID and IS results as reported in the Consistency Models paper [26]. We then apply our boosting algorithm using both a robust classifier and our joint model. The results demonstrate that our proposed approach offers greater benefits compared to refining with a robust classifier alone (i.e., BIGROC [6]). Notably, the proposed training of our joint model leads to significantly improved results. This boost in image quality is observed across both Consistency Training (CT) and Consistency Distillation (CD) methods, and when using either one or two generative steps. In terms of efficiency, one PGD step requires only 0.02 seconds, while a single Consistency (CT or CD) inference step takes 0.55 seconds, making 25 PGD steps comparable in cost to a Consistency inference. This suggests that our approach could be interpreted as providing a continuum trade-off between complexity and perceptual quality. For comparison, [17] used a GAN-based approach to improve CT synthesis with adversarial training, achieving an FID of 10.6, which slightly outperforms BIGROC but falls short of our joint model's performance boost. Note that their training is much more extensive than ours, and much heavier in computational complexity, due to the size of the CT network and the adversarial loss being used.

Table 1: Quantitative comparison of image generation performance on ImageNet 64×64 using Consistency Models and our proposed boosting methods with robust classifiers and joint models.

| Boosting Method | Inference Steps | FID ($\downarrow$) | IS ($\uparrow$) | Precision ($\uparrow$) | Recall ($\uparrow$) |
|---|---|---|---|---|---|
| **Consistency Training** | 1 | 13.00 | 28.83 | 0.71 | 0.41 |
| RN50 Robust Classifier | | 10.71 | 36.38 | 0.70 | **0.47** |
| Wide-RN Non-Robust Joint | | 12.30 | 35.62 | 0.64 | 0.46 |
| RN50 Robust Joint | | 9.60 | 50.80 | 0.76 | 0.39 |
| Wide-RN Robust Joint | | **8.83** | **55.67** | **0.73** | 0.45 |
| **Consistency Training** | 2 | 11.12 | 27.28 | 0.68 | **0.54** |
| RN50 Robust Classifier | | 9.65 | 33.55 | 0.68 | 0.54 |
| Wide-RN Non-Robust Joint | | 11.28 | 33.16 | 0.63 | 0.54 |
| RN50 Robust Joint | | 8.51 | 39.40 | **0.74** | 0.49 |
| Wide-RN Robust Joint | | **7.98** | **46.33** | 0.71 | 0.52 |
| **Consistency Distillation** | 1 | 6.20 | 39.87 | 0.67 | **0.63** |
| RN50 Robust Classifier | | 5.55 | 44.65 | 0.68 | 0.61 |
| Wide-RN Non-Robust Joint | | 6.33 | 45.55 | 0.64 | 0.60 |
| RN50 Robust Joint | | 4.95 | 51.98 | **0.72** | 0.58 |
| Wide-RN Robust Joint | | **4.84** | **58.73** | 0.70 | 0.60 |
| **Consistency Distillation** | 2 | 4.69 | 42.28 | 0.68 | **0.63** |
| RN50 Robust Classifier | | 4.20 | 46.78 | 0.69 | 0.63 |
| Wide-RN Non-Robust Joint | | 4.66 | 46.37 | 0.68 | 0.63 |
| RN50 Robust Joint | | 3.84 | 51.12 | **0.72** | 0.60 |
| Wide-RN Robust Joint | | **3.78** | **55.13** | 0.71 | 0.61 |

## 4.3 Ablation Study

As explained in Section 3.2, the choice of the loss function in Algorithm 2 is crucial. The results in Table 2 illustrate that utilizing only Cross-Entropy loss produces results that are similar to those achieved using BIGROC [6], as expected. The joint model demonstrates substantial discriminative power, which further enhances FID scores when Binary Cross-Entropy loss is applied. In exploring the integration of these two components, we experimented with the loss function proposed in equation 11. Our findings indicate that this approach improves upon Cross-Entropy loss, though it does not consistently surpass Binary Cross-Entropy loss. Consequently, we recommend adopting the loss outlined in Equation 12, which shows an additional improvement in FID by approximately 0.1.

Table 2: FID and IS results with varying loss functions as described in section 3.2.

| Method | Inference Steps | Loss Function | FID ($\downarrow$) | IS ($\uparrow$) |
|---|---|---|---|---|
| **Consistency Training** | 1 | | 13.00 | 28.83 |
| | | $\ell_{CE}$ | 10.78 | 38.97 |
| | | $\ell_{BCE}$ | 9.75 | 44.19 |
| | | $\ell_{CE} - \ell_{BCE}$ | 9.71 | 43.38 |
| | | Our loss | **9.60** | **50.80** |
| **Consistency Training** | 2 | | 11.10 | 27.28 |
| | | $\ell_{CE}$ | 10.26 | 35.89 |
| | | $\ell_{BCE}$ | 8.59 | 39.38 |
| | | $\ell_{CE} - \ell_{BCE}$ | 8.62 | 39.02 |
| | | Our loss | **8.51** | **39.40** |
| **Consistency Distillation** | 1 | | 6.20 | 39.87 |
| | | $\ell_{CE}$ | 5.51 | 47.54 |
| | | $\ell_{BCE}$ | 5.00 | 48.97 |
| | | $\ell_{CE} - \ell_{BCE}$ | 5.10 | 48.63 |
| | | Our loss | **4.95** | **51.98** |
| **Consistency Distillation** | 2 | | 4.69 | 42.28 |
| | | $\ell_{CE}$ | 4.17 | 47.81 |
| | | $\ell_{BCE}$ | 3.89 | 48.72 |
| | | $\ell_{CE} - \ell_{BCE}$ | 4.97 | 48.35 |
| | | Our loss | **3.84** | **51.12** |

We also compare sampling methods — Projected Gradient Descent (PGD) and Stochastic Gradient Langevin Dynamics (SGLD) — as discussed in Section 3.2 and presented in Table 3. Our experiments reveal that SGLD offers advantages over PGD in certain contexts, particularly in terms of sampling diversity and fidelity.

Table 3: Quantitative comparison of our proposed boosting methods using Wide-ResNet with PGD and SGLD sampling techniques applied to Consistency Models.

| Boosting Method | Inference Steps | FID ($\downarrow$) | IS($\uparrow$) | Precision ($\uparrow$) | Recall ($\uparrow$) |
|---|---|---|---|---|---|
| **Consistency Training** | 1 | 13.00 | 28.83 | 0.71 | 0.41 |
| Wide-RN+PGD | | 8.83 | 55.67 | 0.73 | 0.45 |
| Wide-RN+SGLD | | **8.71** | **59.76** | **0.73** | **0.46** |
| **Consistency Training** | 2 | 11.12 | 27.28 | 0.68 | **0.54** |
| Wide-RN+PGD | | 7.98 | 46.33 | 0.71 | 0.52 |
| Wide-RN+SGLD | | **7.64** | **46.83** | **0.71** | 0.53 |
| **Consistency Distillation** | 1 | 6.20 | 39.87 | 0.67 | **0.63** |
| Wide-RN+PGD | | 4.84 | 58.73 | 0.70 | 0.60 |
| Wide-RN+SGLD | | **4.65** | **59.00** | **0.71** | 0.60 |
| **Consistency Distillation** | 2 | 4.69 | 42.28 | 0.68 | **0.63** |
| Wide-RN+PGD | | 3.78 | 55.13 | 0.71 | 0.61 |
| Wide-RN+SGLD | | **3.58** | **56.46** | **0.72** | 0.61 |

To assess the generalization capability of our joint model, we evaluated it on images generated by BigGAN and ADM-G at resolutions of 128×128 and 256×256 pixels (see Table 4). Despite being trained exclusively on images from Consistency Models, our joint model demonstrates strong generalization across different generative models and resolutions. This suggests that our model can enhance images from a variety of generative sources without requiring retraining or fine-tuning for each specific case.

Table 4: Quantitative comparison for various generative models at different resolutions (64×64, 128×128, and 256×256), before and after applying our proposed boosting method using Wide-ResNet with SGLD.

| Boosting Method | FID ($\downarrow$) | IS ($\uparrow$) | Precision ($\uparrow$) | Recall ($\uparrow$) |
|---|---|---|---|---|
| **Adm 64x64** | 2.61 | 46.78 | 0.73 | **0.63** |
| Wide-RN+SGLD | **2.08** | **58.41** | **0.75** | 0.60 |
| **BigGAN 64x64** | 4.06 | 44.94 | 0.79 | **0.48** |
| Wide-RN+SGLD | **3.67** | **62.79** | **0.81** | 0.45 |
| **Iddpm 64x64** | 2.92 | 45.62 | 0.73 | **0.62** |
| Wide-RN+SGLD | **2.25** | **62.26** | **0.76** | 0.59 |
| **Adm-G 128x128** | 2.97 | 141.47 | 0.78 | **0.59** |
| Wide-RN+SGLD | **2.34** | **271.95** | **0.80** | 0.58 |
| **BigGAN 128x128** | 6.02 | 145.83 | 0.86 | 0.34 |
| Wide-RN+SGLD | **5.68** | **236.88** | **0.86** | **0.34** |
| **Adm-G 256x256** | 4.58 | 186.83 | 0.81 | 0.52 |
| Wide-RN+SGLD | **3.17** | **301.06** | **0.83** | **0.53** |
| **BigGAN 256x256** | 7.03 | 202.64 | 0.87 | 0.27 |
| Wide-RN+SGLD | **6.16** | **275.25** | **0.88** | **0.28** |

## 5   Conclusion

This work presents a novel technique for leveraging the perceptually aligned gradients phenomenon for refining synthesized images. Our approach enhances the perceptual quality of images generated by Consistency-based techniques using a robust *joint* classifier-discriminator model. The dual functionality of our model, which serves as both classification and energy-based discrimination, leads to significant improvements in image fidelity.

**Limitations:** In this work we have used the RN50 and Wide-RN 50-2 architectures for the joint model, as commonly employed in robust classification tasks. We note that this architectures are relatively simple and small, compared to the more complex ones proposed in recent years. As a consequence, our study is unavoidably constrained by the capabilities of the chosen models. Another limitation in our work is the reliance of our training on Consistency generated images only. Expanding the training-set to include images from additional generative models could further improve the overall boosting effect that we document in this paper.

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

## A  EBM via Classification

A classifier $f_\theta : \mathbb{R}^d \to \mathbb{R}^C$ is a function that maps data points into logit vectors that can parameterize a posterior distribution of the form

$$p_\theta(c|x) = \frac{exp(f_\theta^c(x))}{\sum_{c'=1}^{C} exp(f_\theta^{c'}(x))} = \frac{exp(f_\theta^c(x))}{N_\theta(x)}, \tag{14}$$

where we have defined the normalizing factor by $N_\theta(x) = \sum_{c'=1}^{C} exp(f_\theta^{c'}(x))$. Axiomatically, let us use the logits of the classifier, as is, in order to define a joint probability $p_\theta(x, c)$:

$$p_\theta(x, c) = \frac{exp(f_\theta^c(x))}{N_\theta}, \tag{15}$$

where $N_\theta$ is yet another normalizing factor. Note that while the above two equations share the same nominator, their denominators are markedly different, as $N_\theta$ normalizes over all $c$ and $x$, and thus $N_\theta = \int_x N_\theta(x)p(x)dx$.

By marginalizing the joint probability $p_\theta(x, c)$ over $c$, and assuming a uniform probability for the classes, $p(c) = 1/C$ for all $c = [1, 2, \ldots, C]$, one can obtain the expression for $p_\theta(x)$:

$$p_\theta(x) = \frac{\sum_{c=1}^{C} exp(f_\theta^c(x))}{C \cdot N_\theta}. \tag{16}$$

However, an EBM probability is defined generally by

$$p_\theta(x) = \frac{exp(-E_\theta(x))}{Z_\theta}, \tag{17}$$

and therefore, by matching terms in the above two equations we get

$$E_\theta(x) = -log \left[ \sum_{c=1}^{C} exp(f_\theta^c(x)) \right], \tag{18}$$

as suggested in Section 2.3. Note that these derivations enable us to construct $p_\theta(c|x) = \frac{p_\theta(x,c)}{p_\theta(x)}$. When doing so, the normalizing factor cancels out, and we obtain the regular softmax.

## B  Challenges of EBMs

EBMs often require careful tuning of their learning parameters, as the energy landscape they model can be complex and difficult to navigate. This complexity can lead to issues with training stability, where the model may fail to converge or converge to undesirable local minima, resulting in poor generalization of new data. Additionally, EBMs are sensitive to the choice of the energy function. An inadequately defined energy function can lead to a model that either assigns high probability to non-data-like samples or fails to capture the diversity of the dataset, thereby affecting the quality of the generated images. The training objective usually used in EBMs aims to adjust $p_\theta(x)$ given in equation (4) such that the model's energy surface aligns with the data distribution, the loss function often employed is based on minimizing the difference between the energy of the real data and the expected energy of the model distribution. The objective is typically formulated as:

$$\mathcal{L}(\theta) = \mathbb{E}_{x \sim p_{data}}[E_\theta(x)] - \mathbb{E}_{x \sim p_\theta}[E_\theta(x)] \tag{19}$$

Where $E_\theta$ is the energy function and $\theta$ is the model parameters. This expression is often approximated using sampling techniques like Markov Chain Monte Carlo (MCMC) because directly computing expectations under $p_\theta$ is computationally infeasible due to the unknown partition function $Z(\theta)$.

EBMs require extensive computational resources, particularly due to the need for sampling during training. Sampling methods such as Markov Chain Monte Carlo (MCMC) are computationally expensive and slow, especially when dealing with high-dimensional data like images and the ImageNet dataset in particular. The complexity and variety within ImageNet demand a model that can capture a vast range of features and variations, pushing the limits of an EBM's capacity to model such diverse data effectively.

## C  EBMs and Adversarial Training

The proposed loss in equation (7) is closely related to GAN's training objective, but can also be leveraged for an energy-based training. Both the EBM loss function (19) and the proposed training objective aim to minimize the energy (or increase the likelihood) of observed data points and maximize the energy (or decrease the likelihood) of generated data. In EBMs, this is done by reducing the energy of real data samples and increasing it for generated samples.

In the logistic regression-based approach that we take in this work, the log-likelihood of the real data under the model (represented by $p_\theta(x_{real}|c) = -log\left(\sigma(f_\theta^c(x_{real}))\right)$) is minimized, while the log-likelihood of the generated data (represented by $p_\theta(x_{gen}|c) = -log(1 - \sigma(f_\theta^c(x_{gen})))$) is maximized. In other words, the energy we use for real/fake is the magnitude of the $c$ logit itself, while its normalized value (after soft-max) is the classification probability.

## D  Supplementary Results

In the following figures we present qualitative results demonstrating that the "boosted" results indeed look better perceptually. The enhancement is visible on the edges and textures and leads to sharper and more high-contrast images, creating a more perceptually pleasing outcome. Note that we recommend zooming in the images to see the changes clearly.

For reproducibility, we elaborate on the amount of steps and the step size used for boosting each generative model in Table 5.

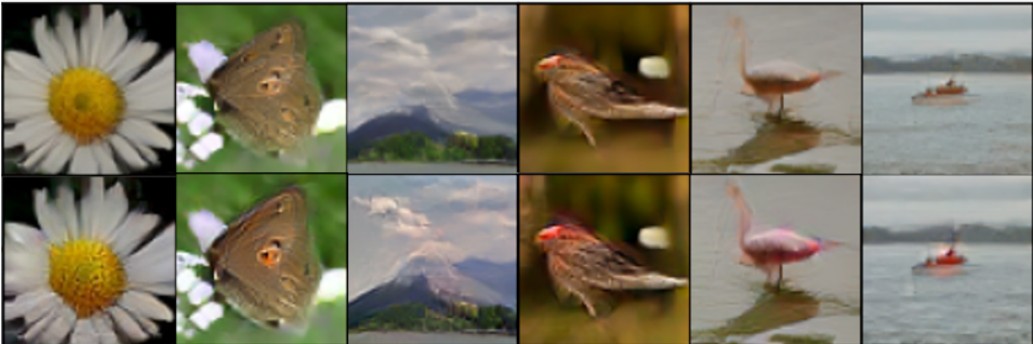

Figure 1: Top: Images generated by Consistency Models trained on ImageNet 64x64. Bottom: Refined images by applying our algorithm.

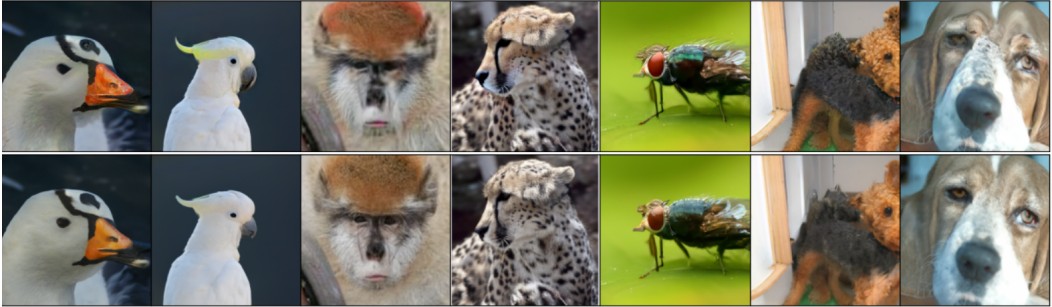

Figure 2: Top: Images generated by Guided Diffusion trained on ImageNet 256x256. Bottom: Refined images by applying our algorithm.

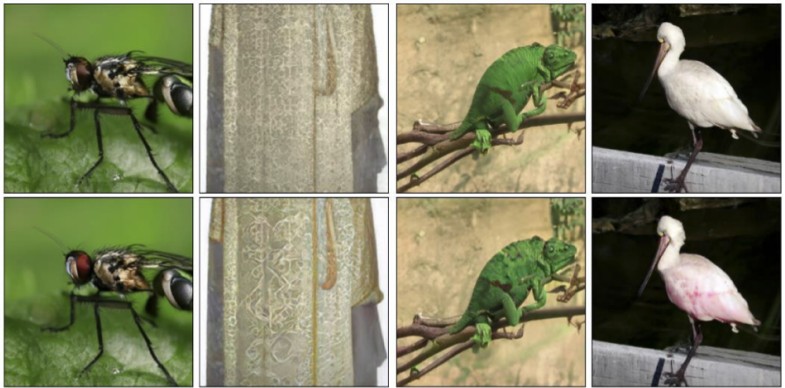

Figure 3: Top: Images generated by Guided Diffusion trained on ImageNet 256x256. Bottom: Refined images by applying our algorithm.

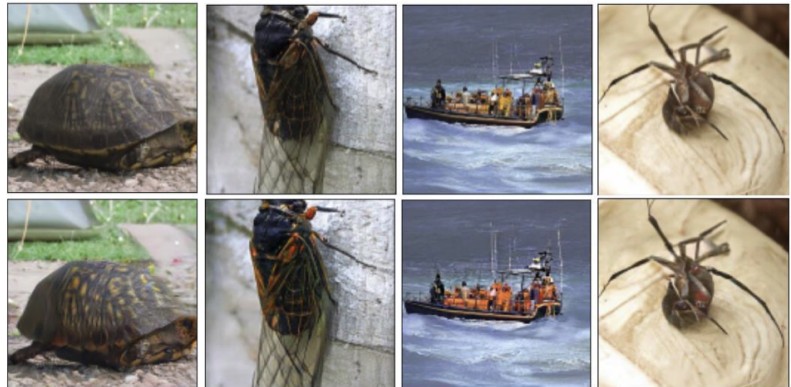

Figure 4: Top: Images generated by Guided Diffusion trained on ImageNet 128x128. Bottom: Refined images by applying our algorithm.

Table 5: Hyper-parameters used in algorithm 2.

| Generative Method | Inference Steps | PGD Steps | Step Size |
|---|---|---|---|
| **Consistency Training** | 1 | 35 | 0.1 |
| **Consistency Training** | 2 | 25 | 0.1 |
| **Consistency Distillation** | 1 | 15 | 0.15 |
| **Consistency Distillation** | 2 | 7 | 0.14 |
| **Admnet [4]** | 250 | 7 | 0.14 |
| **BigGAN [2]** | 1 | 15 | 0.1 |
| **Iddpm [22]** | 1000 | 7 | 0.14 |

