# OpenReview forum: "Enhancing Consistency-Based Image Generation via Adversarialy-Trained Classification and Energy-Based Discrimination"
_NeurIPS.cc/2024/Conference — NeurIPS 2024 poster_

### Official Review · Reviewer_zvWf · 2024-07-10

**Soundness:** 3
**Presentation:** 2
**Contribution:** 2
**Rating:** 5
**Confidence:** 3

**Summary:**

The paper introduces a post-processing method using a joint classifier-discriminator model to imporve the generation quality of consistency models. This model is trained adversarially, with the robust classifier providing gradient based on class assignment and the discriminator score. This post-processing method consistently improves the quality of the generated images in both single-step and double-step cases, resulting in better FID scores for both consistency distillation and consistency training.

**Strengths:**

- The method has connections, and it is possible to unify the Robust Classifier and Energy-based Models with consistency models.

- The proposed method is efficient and easy to plug-and-play in consistency distillation and consistency training.

- The paper is overall easy to read and well organized.

**Weaknesses:**

- Although the method seems to be connected to the Robust Classifier and Energy-based Models, the authors did not give comprehensive theoretical derivation of the connections. The resulted loss is simply linear combination of binary CE and CE.

- Is the proposed Robust Joint-Model specifically useful for consistency model? If so the authors may need to give analysis why this is the case. If not, it worth to study how the proposed method applicable with other generative models.

- How does the radius $\epsilon$ affects the final performance? Also, how do the other hyper-parameters affect the model performance? The authors should give ablation studies regarding these parts.

- From the empirical results in Table 3, the improvement of the final loss is marginal compared to using CE and BCE only sometimes.

Some minors:

- line 124: algorithm (missed) 1.

**Questions:**

Please see the Weakness.

**Limitations:**

The authors have discussed the limitations and potential negative societal impact of their work.

---

> ### Author Rebuttal · Authors · 2024-08-06
>
> We thank the reviewer for the detailed review and insightful questions. These constructive comments provide valuable perspectives to enhance our work. Additionally, we are grateful for your recognition of the efficiency of our method.
> Below, we address the reviewer’s specific concerns:
>
>
> >Although the method seems to be connected to the Robust Classifier and Energy-based Models, the authors did not give comprehensive theoretical derivation of the connections. The resulted loss is simply linear combination of binary CE and CE.
>
> We appreciate the reviewers' comment and the opportunity to clarify the connections between our method and Robust Classifiers as well as Energy-based Models. Below, we address the specific concerns regarding the theoretical derivation of these connections and the composition of the resultant loss.
> Robust classifiers are integral to our approach as they provide Perceptually Aligned Gradients (PAG), which are crucial for guiding the refinement of synthesized images. The iterative process of modifying images to maximize the conditional probability of a target class relies on the robust classifier's ability to produce meaningful and perceptually consistent gradients. These gradients ensure that changes to the image are visually aligned with the intended class, thereby improving image quality.
> The discriminator component of our joint model serves an analogous function to an Energy-based Model. It uses the softmax values from the classifier to evaluate the proximity of the input image to the targeted data manifold. The energy function  can be derived from the classifier’s logits as shown in equation 5: \
>  $E_\theta(x)=-log\left[\sum_{c=1}^C exp\left(f_\theta^c(x)\right) \right]$.\
> This formulation is aligned with the principles of Energy-based Models, where lower energy values correspond to higher probabilities of being real images, further justifications can be found in appendix A and C.
> We acknowledge the reviewer's observation regarding the resultant loss being a linear combination of binary cross-entropy (BCE) and cross-entropy (CE). This design integrates the classification and discrimination tasks into a unified objective. The BCE component is inspired by the familiar GAN loss, focusing on distinguishing real from fake images, while the CE component ensures accurate class alignment. This dual loss structure is essential for harnessing the strengths of both robust classification and energy-based discrimination. We further analyze the relation between the robust classifier and energy-based discrimination empirically and add additional ablation study regarding the importance of the energy-based components.
>
> > Is the proposed Robust Joint-Model specifically useful for consistency model? If so the authors may need to give analysis why this is the case. If not, it worth to study how the proposed method applicable with other generative models.
>
> While our primary focus has been the enhancement of consistency models, we acknowledge that the principles underlying our Robust Joint-Model are general. Indeed, several additional experiments that are reported in the paper show that other generative models (ADM-G, BigGAN,...) may benefit from our post-processing algorithm. Furthermore, in all these experiments, we did not retrain the joint classifier-discriminator model, implying that while it has been trained on consistency images, it serves well other generative images. This zero-shot approach has also been practiced in new experiments that we now add, in order to include 128x128 and 256x256 images, generated by bigGAN and ADM-G.
>
> >How does the radius $\epsilon$ affects the final performance? Also, how do the other hyper-parameters affect the model performance? The authors should give ablation studies regarding these parts.
>
> We will add ablation studies regarding the effect of the radius on the performance in sampling and training, as this was the only parameter we tuned during training, the rest had negligible influence on the final results. The hyper-parameters we chose are specified in section 4.1.
> As noted in the paper, for real images, the PGD attack was set with a radius of 1.0, chosen after experimenting with different values, as described in the table below.
>
> | Method | $\varepsilon$ | FID | IS |
> | ---- | :---: | --- | --- |
> |CT (1 step) | 1.0 | 9.60 | 50.80 |
> |    | 2.0 | 9.98 | 40.96 |
> |    | 3.0 | 10.23 | 39.12 |
> | CD (1 step) | 1.0 | 4.95 | 51.98 |
> |      | 2.0 | 5.13 | 47.28 |
> |       | 3.0 | 5.24 | 46.25 |
>
> We will provide extended results in our revised manuscript. \
> Regarding the sampling process, we have provided the PGD parameters used in Table 4 (Appendix D) of our paper. We will include further analysis of the results obtained with different numbers of steps in our revised version.
>
> > From the empirical results in Table 3, the improvement of the final loss is marginal compared to using CE and BCE only sometimes.
>
> In Table 3 we show the results for $\ell_{CE}$ alone, $\ell_{BCE}$ alone, the two together, and our loss that incorporates the above with a target value for the BCE loss - the last one is referred to in the table as “Our Loss”.
>
> As for the improvements between these options:
> 1. Our Binary Cross-Entropy (BCE) component is derived from the logits produced by the classifier. This means that it inherently utilizes the knowledge distilled from the classifier through the Cross-Entropy (CE) component. Consequently, the BCE component incorporates significantly more information than the CE component alone. This synergy between BCE and CE ensures that our model benefits from a comprehensive understanding of both the classification and discrimination tasks.
> 2. The proposed final loss achieves an additional improvement of up to 0.15 FID over the plain $\ell_{CE}-\ell_{BCE}$ loss. Although this is not a major improvement, we should note that it does not require additional computational resources at all, compared to the simpler alternative.

---

> > ### Comment · Reviewer_zvWf · 2024-08-10
> >
> > I appreciate the time and effort the authors have taken to address the concerns. The clarifications have addressed my questions and I have increased my score. Please make sure to incorporate these discussions in the final revision.

---

### Official Review · Reviewer_JUbG · 2024-07-11

**Soundness:** 3
**Presentation:** 3
**Contribution:** 3
**Rating:** 6
**Confidence:** 3

**Summary:**

This paper proposes a post-processing method to improve the generated image quality of consistency models. The method mainly includes a joint classifier-discriminator, which contains an adversarial training objective and a gradual training approach. Experimental results show the effectiveness of the proposed method.

**Strengths:**

(++) The method can be seamlessly integrated into generative models, facilitating applications.

(+) The writing is easy to follow.

**Weaknesses:**

(--) The evaluations are limited. Only ImageNet at resolution 64x64 is utilized, which could not be representative since the development trend is high-resolution generation. Datasets that contain larger images should be considered, such as LSUN and FFHQ with resolution larger than 256x256. Meanwhile, more metrics, such as the precision and recall, are also worth considering for better evaluation about the data distribution.

Minor issue:
1. A typo at line 55: We show a ... improvements ...
2. The case rules of the journal or conference names in the reference are inconsistent sometimes.

**Questions:**

Please refer to Weaknesses.

**Limitations:**

The limitations have been well discussed.

---

> ### Author Rebuttal · Authors · 2024-08-06
>
> Dear reviewer, we sincerely appreciate the time and effort you dedicated to reviewing our paper. Your thoughtful and constructive feedback has been invaluable in highlighting the strengths of our work.
> We address your concerns below:
>
> >The evaluations are limited. Only ImageNet at resolution 64x64 is utilized, which could not be representative since the development trend is high-resolution generation. Datasets that contain larger images should be considered, such as LSUN and FFHQ with resolution larger than 256x256. Meanwhile, more metrics, such as the precision and recall, are also worth considering for better evaluation about the data distribution.
>
> We appreciate the important comment. In response, we have conducted additional experiments on two high-resolution versions of ImageNet (128 and 256), using both RN-50 and Wide ResNet 50-2. \
> The CT/CD models we have access to are limited to images from ImageNet of size 64x64. However, while our method primarily focuses on Consistency based image generation, it can be applied on images synthesized by any generative model. Therefore, we can evaluate our model on other generative methods at different resolutions. Furthermore, this test can be done, even without any further training or adjustments, which is exactly the experiment we have conducted and reported herein. We have applied our method to images synthesized by BigGAN and Guided Diffusion (ADM-G) at resolutions of 128x128 and 256x256. Our findings are reported in Table 3 in the attached file. \
> Please note that our work refers to data that carries labels for classification (e.g. ImageNet), and thus face images as in FFHQ are not currently covered by our approach. \
> Referring to the LSUN dataset (256x256), we applied our method as is (no re-training) and the preliminary results are reported in the table below:
>
> | Method | Dataset | FID |
> | ---- | --- | --- |
> | ADM-G | LSUN-cats | 5.57 |
> | Robust Joint-Model | LSUN-cats | **4.97** |
> | DDPM | LSUN-bedroom | 4.89 |
> | Robust Joint-Model |  LSUN-bedroom | **4.75** |
> | ADM-G | LSUN-horses | 2.95 |
> | Robust Joint-Model |  LSUN-horses | **2.77** |
>
> We will also include precision and recall metrics in all the relevant tables in the paper.
>
> >A typo at line 55: We show a ... improvements ...
> >The case rules of the journal or conference names in the reference are inconsistent sometimes.
>
> Thank you for this correction; we will include this in our revised version.

---

> > ### Comment · Reviewer_JUbG · 2024-08-10
> >
> > Thank you for your response. I recommend including these results in the paper, or the Appendix if they exceed the page limit. There are no more questions.

---

### Official Review · Reviewer_4wrH · 2024-07-12

**Soundness:** 3
**Presentation:** 3
**Contribution:** 3
**Rating:** 7
**Confidence:** 3

**Summary:**

The paper proposes a method to mitigate the quality degradation of  images generated by consistency-based generative models compared to diffusion models.

The main idea is to adapt the generated image based on the logits of an adversarial real/synthetic image discriminator to maximize the image resemblance to real images. A single classifier is trained jointly for both discrimnation between real/synthesized images and for assignment to designated categories.
Optimization in the generative model's output space makes the method more efficient than methods based on updating the model parameters.

The method is evaluated on the ImageNet 64x64 dataset, showing performance gains in FID and IS over consistency-based models.

**Strengths:**

1. The method is novel, simple (easy to understand and implement) and outperforms baselines on the evaluation dataset. It has the potential to inspire further research in this direction.
2. The paper is easy to follow, all the methodology is well justified and well explained, design choices such as the loss function are supported by  ablations
3. Both CT and CD, with different number of inference steps are evaluated

**Weaknesses:**

1. Evaluation on a single dataset with small resolution is not sufficient to judge the performance of genrative methods.
2. Very limited qualitative results

See questions for more details on both.

**Questions:**

### Weakness 1:
1. Other datasets such as ImageNet with higher resolution, COCO or flicker faces would help to judge the method’s performance and generalizability.

### Weakness 2:
2. While the method outperforms baselines in quantitative evaluation, more visualizations should be provided, especially since the method is about generative models. I would recommend including  a dataset like Flicker Faces for visualizations since human perception is sensitive to artefacts in the human face.

### Typos.:
3. Line 93 - 'in two main ways []' - missing citations
4. Line 124 – something went wrong at the beginning of the line, '1' should be 'Alg. 1' for better readability, also maybe the sentence should end after Alg. 1?

### Presentation:
5. While the paper is clearly written, it would be helpful to have a figure overview of the pipeline, rather than just equations.

**Limitations:**

Limitations are addressed in a dedicated section.

---

> ### Author Rebuttal · Authors · 2024-08-06
>
> Dear reviewer, we sincerely appreciate the time and effort you dedicated reviewing our paper. Your thoughtful and constructive feedback has been invaluable in highlighting the strengths of our work. Your insightful reviews and acknowledgment of our contributions inspire us to continue refining our work in this field. We address your concerns below:
>
> > Other datasets such as ImageNet with higher resolution, COCO or flicker faces would help to judge the method’s performance and generalizability.
>
> We appreciate the important comment. In response, we have conducted additional experiments on two high-resolution versions of ImageNet (128 and 256), using both RN-50 and Wide ResNet 50-2.
> The CT/CD models we have access to are limited to images from ImageNet of size 64x64. However, while our method primarily focuses on Consistency based image generation, it can be applied on images synthesized by any generative model. Therefore, we can evaluate our model on other generative methods at different resolutions. Furthermore, this test can be done, even without any further training or adjustments. We have applied our method to images synthesized by BigGAN and ADM-G at resolutions of 128x128 and 256x256. Our findings are reported in Table 3 in the attached file.
>
> We also ran experiments on the LSUN dataset (256x256). We applied our method as is (no re-training) and the preliminary results are reported in the table below:
>
> | Method | Dataset | FID |
> | ---- | --- | --- |
> | ADM-G | LSUN-cats | 5.57 |
> | Robust Joint-Model | LSUN-cats | **4.97** |
> | DDPM | LSUN-bedroom | 4.89 |
> | Robust Joint-Model |  LSUN-bedroom | **4.75** |
> | ADM-G | LSUN-horses | 2.95 |
> | Robust Joint-Model |  LSUN-horses | **2.77** |
>
> > While the method outperforms baselines in quantitative evaluation, more visualizations should be provided, especially since the method is about generative models. I would recommend including a dataset like Flicker Faces for visualizations since human perception is sensitive to artefacts in the human face.
>
> We appreciate the comment regarding qualitative results. We will include more figures in the paper to illustrate the gain in visual quality. Some of those illustrations can be found in the attached file - figures 1,2.
> Please note that our work refers to data that carries labels for classification (e.g. ImageNet), and thus face images are not currently covered by our approach.
>
> >Line 93 - 'in two main ways []' - missing citations
> >Line 124 – something went wrong at the beginning of the line, '1' should be 'Alg. 1' for better readability, also maybe the sentence should end after Alg. 1?
>
> Thank you for the corrections; we will include them in our revised version.
>
> >While the paper is clearly written, it would be helpful to have a figure overview of the pipeline, rather than just equations.
>
> We will attach an illustration of both the training and the inference parts of our proposed approach in the revised version of our manuscript.

---

> > ### Comment · Reviewer_4wrH · 2024-08-11
> >
> > Thank you for the response and effort! I am increasing my score based on the additional experiments.
> >
> > My biggest remaining concern is the qualitative results; they mostly seem limited to color enhancements with maybe the exception of the robe image in Figure 1.
> > I would still consider qualitative results with at least animal faces.

---

### Official Review · Reviewer_Jjs5 · 2024-07-14

**Soundness:** 3
**Presentation:** 3
**Contribution:** 2
**Rating:** 5
**Confidence:** 4

**Summary:**

The manuscript proposes a method for refining consistency-based image generation using a joint-trained classifier-discriminator. The method helps to improve the quality of the fast-sampling consistency models.

**Strengths:**

1. The paper is well written

2. the performance is significant

**Weaknesses:**

1. The motivation of the work is not clear. The main motivation of the work is to postprocess generated images to improve perceptual quality (line 36-38). In lines 39 - 41, the authors mention the adversarial attacks as the main reason why the perceptual quality is low. There is a gap between these two problems. I agree that an adversarial problem would cause poor alignment, but this phenomenon happens to other models as well, not only the consistency-based model.

2. Furthermore, the proposed method does not only focus on the robustness problem (only in section 3.1.2). The motivation behind joint classifier-discriminator is not clear. Although this scheme is very popular in many generative models, it is not easy to point out its motivation in the context of the manuscript. From line 170 to 175, there is a mention about the drawback of EBMs, yet it is not clear how classifier-discriminator training would solve it.

3. The analysis is lacked. From line 170 to 175, the authors mention about the sensitivity of EBM, instability and poor generalization. However, no evidence is provided.

4. The qualitative improvement is limited and hard to observe. This raises the question whether the method is effective or not.

**Questions:**

see weaknesses.

---

> ### Author Rebuttal · Authors · 2024-08-06
>
> We thank the reviewer for the detailed review and insightful questions. These constructive comments provide valuable perspectives to enhance our work. Additionally, we are grateful for your recognition of the significant performance of our method.
> Below, we address the reviewer’s specific concerns:
>
> > The motivation of the work is not clear. The main motivation of the work is to postprocess generated images to improve perceptual quality (line 36-38). In lines 39 - 41, the authors mention the adversarial attacks as the main reason why the perceptual quality is low. There is a gap between these two problems. I agree that an adversarial problem would cause poor alignment, but this phenomenon happens to other models as well, not only the consistency-based model.
>
> We apologize for the lack of clarity in reference to this point. As correctly noted, the primary motivation of our work is to post-process generated images (mainly those created by Consistency models) to enhance their perceptual quality. This is achieved by training yet another network -- a joint classifier-discriminator model. As explained in the paper, this additional model must be trained adversarially in order to enable the desired boost in image quality. The joint classifier-discriminator model is leveraged via the gradients of this network, computed in order to apply an iterative PGD procedure. Please note that in all this process, we do not alter the generative model itself.
>
> >Furthermore, the proposed method does not only focus on the robustness problem (only in section 3.1.2). The motivation behind joint classifier-discriminator is not clear. Although this scheme is very popular in many generative models, it is not easy to point out its motivation in the context of the manuscript.
>
> The motivation behind the use of a  joint classifier-discriminator model is the following:
> 1. Given a classifier, its gradients can be used for pushing the input image towards better matching its referenced class. However, prior work [1] has shown that this is effective only if the classifier is robust, i.e. trained adversarially.
> 2. Given an Energy-Based-Model (which we refer to as a `discriminator’), its gradients can be used for pushing the input image towards the image manifold, i.e. being aligned with the distribution of real images.
> 3. Merging the above two into a single network, and robustifying both the classifier and the discriminator objectives, we end up with a single joint model which can guide the image improvement much better, which is the essence of this work.
>
> To validate the effectiveness of our joint classifier-discriminator model, we have included an ablation study [Table 2 in the paper] which demonstrates that enhancing images using only a robust classifier is not as powerful as using our joint model.
>
> >From line 170 to 175, there is a mention about the drawback of EBMs, yet it is not clear how classifier-discriminator training would solve it.
>
> > The analysis is lacked. From line 170 to 175, the authors mention about the sensitivity of EBM, instability and poor generalization. However, no evidence is provided.
>
> EBMs and JEMs are known to be unstable during training. The work reported in [2] highlights these challenges in their limitations section. Similarly, [3] addresses the difficulties of training EBMs. Inspired by these papers, we propose utilizing robustness and deterministic sampling to overcome those issues. \
> More specifically, [2] suggests optimizing the energy portion of the classifier using Stochastic Gradient Langevin Dynamics (SGLD) during training. However, they reported encountering instabilities and convergence issues. ⁠In contrast, we found that using Projected Gradient Descent (PGD) along with our alternative loss objective (+ robustness) significantly enhances stability.
>
> To further investigate the properties of SGLD, we explored the option of replacing PGD during inference and training time. In the training process we found that optimizing with SGLD lacks the robustness achieved with PGD and suffers from difficulties to converge; at inference though it surpasses PGD greatly. These new results will be incorporated into the revised version of the paper.
>
> >The qualitative improvement is limited and hard to observe. This raises the question whether the method is effective or not.
>
> We appreciate the comment regarding qualitative results. In response, we will include more figures in the paper to illustrate the gain in visual quality. Some of those illustrations can be found in the attached file - figures 1,2. \
> Please note that while this improvement is mild in small (64x64) images, we have included new results that refer to larger (128x128 and 256x256) images, and there the benefit is clearly seen.
>
>
> We hope these additional details and clarifications address the reviewers' concerns.
>
>
> [1] Ganz, Roy, et al. "BIGRoC: Boosting Image Generation via a Robust Classifier." Transactions on Machine Learning Research. 2021. \
> [2] Grathwohl, Will, et al. "Your classifier is secretly an energy based model and you should treat it like one." International Conference on Learning Representations. 2019.‏\
> [3] Song, Yang, et al. "How to Train Your Energy-Based Models." arXiv e-prints 2021.‏

---

> > ### Comment · Reviewer_Jjs5 · 2024-08-12
> > **Thanks for the response**
> >
> > The rebuttal has addressed my concerns.
> >
> > I will increase the score.
> >
> > Best regards,

---

### Author Rebuttal · Authors · 2024-08-06

We appreciate the reviewers' constructive feedback and their recognition of our method's novelty and simplicity (R2), as well as its potential for integration into various generative models (R2, R3, R4). We have carefully considered the main issues raised by the reviewers and have taken steps to address them.

We agree that providing more extensive quantitative and qualitative experiments is crucial for establishing the effectiveness of our approach. To this end, we have conducted additional experiments on two high-resolution versions of ImageNet (128 and 256), using both RN-50 and Wide ResNet 50-2. These results are reported in the attached file. \
Our findings indicate that the Wide ResNet 50-2 outperforms the RN-50 and the baselines, leading to substantial quantitative improvements, particularly in enhancing low-level details in high-resolution images. These improvements are reflected in the additional qualitative evaluations presented in the attached file.

Additionally, we explored the effects of Stochastic Gradient Langevin Dynamics (SGLD), which are typically used in the Energy-Based Models (EBMs) training and inference. Our findings indicate that replacing the PGD algorithm in the sampling process (described in section 3.2) with SGLD is beneficial and achieves significantly better results. However, training the joint model with SGLD led to poor performance. These results support our assumption that adversarial training is crucial to the process. We will include a detailed analysis in our revised version of the paper.

---

### Decision · Program_Chairs · 2024-09-25

**Decision:**

Accept (poster)

**Comment:**

The paper proposes enhancing the realism of images from consistency-based generative models by using an adversarial discriminator's logits to optimize the output space, improving efficiency over parameter adjustments. A single classifier is trained for both real/synthetic discrimination and category assignment. Overall, the reviewers are satisfied with the authors' response. A minor concern is that the qualitative results mainly show color enhancements, with limited examples of significant improvements, such as detailed animal faces. It is better for the authors to clarify this issue further.